# RetroXpert: Decompose Retrosynthesis Prediction Like A Chemist

**Chaochao Yan**[*†]
University of Texas at Arlington
chaochao.yan@mavs.uta.edu

**Qianggang Ding** [*†]
Tsinghua University
dqg18@mails.tsinghua.edu.cn

**Peilin Zhao**
Tencent AI Lab
masonzhao@tencent.com

**Shuangjia Zheng** [†]
Sun Yat-sen University
zhengshj9@mail2.sysu.edu.cn

**Jinyu Yang** [†]
University of Texas at Arlington
jinyu.yang@mavs.uta.edu

**Yang Yu**
Tencent AI Lab
kevinyyu@tencent.com

**Junzhou Huang**
University of Texas at Arlington
jzhuang@uta.edu

## Abstract

Retrosynthesis is the process of recursively decomposing target molecules into available building blocks. It plays an important role in solving problems in organic synthesis planning. To automate or assist in the retrosynthesis analysis, various retrosynthesis prediction algorithms have been proposed. However, most of them are cumbersome and lack interpretability about their predictions. In this paper, we devise a novel template-free algorithm for automatic retrosynthetic expansion inspired by how chemists approach retrosynthesis prediction. Our method disassembles retrosynthesis into two steps: i) identify the potential reaction center of the target molecule through a novel graph neural network and generate intermediate synthons, and ii) generate the reactants associated with synthons via a robust reactant generation model. While outperforming the state-of-the-art baselines by a significant margin, our model also provides chemically reasonable interpretation.

## 1 Introduction

Retrosynthesis of the desired compound is commonly constructed by recursively decomposing it into a set of available reaction building blocks. This analysis mode was formalized in the pioneering work [1, 2] and now have become one of the fundamental paradigms in the modern chemical society. Retrosynthesis is challenging, in part due to the huge size of the search space. The reported synthetic-organic knowledge consists of in the order of $10^7$ reactions and compounds [3]. On the other hand, the incomplete understanding of the reaction mechanism also increases the difficulty of retrosynthesis, which is typically undertaken by human experts. Therefore, it is a subjective process and requires considerable expertise and experience. However, molecules may have multiple possible retrosynthetic routes and it is challenging even for experts to select the most appropriate route since the feasibility of a route is often determined by multiple factors, such as the availability of potential reactants, reaction conditions, reaction yield, and potential toxic byproducts.

---

[*]Both authors contribute equally to the work.

[†]This work is done when Chaochao Yan, Qianggang Ding, Shuangjia Zheng, and Jinyu Yang work as interns at Tencent AI Lab.

In this work, we focus on the single-step version (predict possible reactants given the product) of retrosynthesis following previous methods [4, 5, 6]. Our method can be decomposed into two sub-tasks [1, 7]: i) *Breaking down* the given target molecule into a set of **synthons** which are hypothetical units representing potential starting reactants in the retrosynthesis of the target, and ii) *Calibrating* the obtained synthons into a set of reactants, each of which corresponds to an available molecule.

Various computational methods [8, 9, 10, 11, 12, 4, 13, 14, 5, 6, 15, 16] have been developed to assist in designing synthetic routes for novel molecules, and these methods can be broadly divided into two template-based and template-free categories. Template-based methods plan retrosynthesis based on hand-encoded rules or reaction templates. Synthia (formerly Chematica) relies on hand-encoded reaction transformation rules [11], and it has been experimentally validated as an efficient software for retrosynthesis [17]. However, it is infeasible to manually encode all the synthesis routes in practice considering the exponential growth in the number of reactions [14]. Reaction templates are often automatically extracted from the reaction databases and appropriate templates are selected to apply to the target [12, 13, 14, 5]. The key process of these approaches is to select relevant templates for the given target. An obvious limitation is that these methods can only infer reactions within the chemical space covered by the template database, preventing them from discovering novel reactions [18].

On the other hand, template-free methods [4, 6, 15] treat the retrosynthesis as a neural machine translation problem, since molecules can be represented as SMILES [3] strings. Although simple and expressive, these models do not fit into the chemists' analytical process and lack interpretability behind their predictions. Besides, such approaches fail to consider rich chemistry knowledge within the chemical reactions. For example, the generation order of reactants is undetermined in [4, 6, 15] since they ignore the correlation between synthons and reactants, resulting in slower and inferior model convergence. Similar to our method, the concurrent work G2Gs [16] also presents a decomposition and generation two-step framework. G2Gs proposes to incrementally generate reactants from the associated synthons with a variational graph translation model. However, G2Gs can predict at most one bond disconnection which is not universal. Besides, G2Gs independently generates multiple reactants, which ignores the relationship between multiple reactants.

To overcome these challenges, inspired by the expert experience from chemists, we devise a two-step framework named as RetroXpert (**Retro**synthesis e**Xpert**) to automate the retrosynthesis prediction. Our model tackles it in two steps as shown in Figure 1. Firstly, we propose to identify the potential reaction center within the target molecule using a novel Edge-enhanced Graph Attention Network (EGAT). The reaction center is referred to as the set of bonds that will be disconnected in the retrosynthesis process. Synthons can be obtained by splitting the target molecule according to the reaction center. Secondly, the Reactant Generation Network (RGN) predicts associated reactants given the target molecule and synthons. Different from previous methods [4, 6, 15], the reactant generation order can be uniquely decided in our method, thanks to the intermediate synthons. What is more, we notice that the robustness of the RGN plays an important role. To robustify the RGN, we propose to augment the training data of RGN by incorporating unsuccessful predicted synthons. Our main contributions can be summarized as follows:

1) We propose to identify the potential reaction center with a novel Edge-enhanced Graph Attention Network (EGAT) which is strengthened with chemical knowledge.

2) By splitting the target molecule into synthons, the RGN is able to determine the generation order of reactants. We further propose to augment training data by introducing unsuccessfully predicted synthons, which makes RGN robust and achieves significant improvement.

3) On the standard USPTO-50K dataset [19], our method achieves 70.4% and 65.5% Top-1 accuracy when w/ and wo/ reaction type, respectively, which outperforms SOTA accuracy 63.2% (w/) and 52.6% (wo/) reported in [5] by a large margin.

## 2 Methodology

Given a molecule graph **G** with $N$ nodes (atoms), we denote the matrix representation of node features as $X \in \mathbb{R}^{N \times M}$, the tensor representation of edge features as $E \in \mathbb{R}^{N \times N \times L}$, and the adjacency matrix as $A \in \{0, 1\}^{N \times N}$. $M$ and $L$ are feature dimensions of atoms and bonds, respectively. We

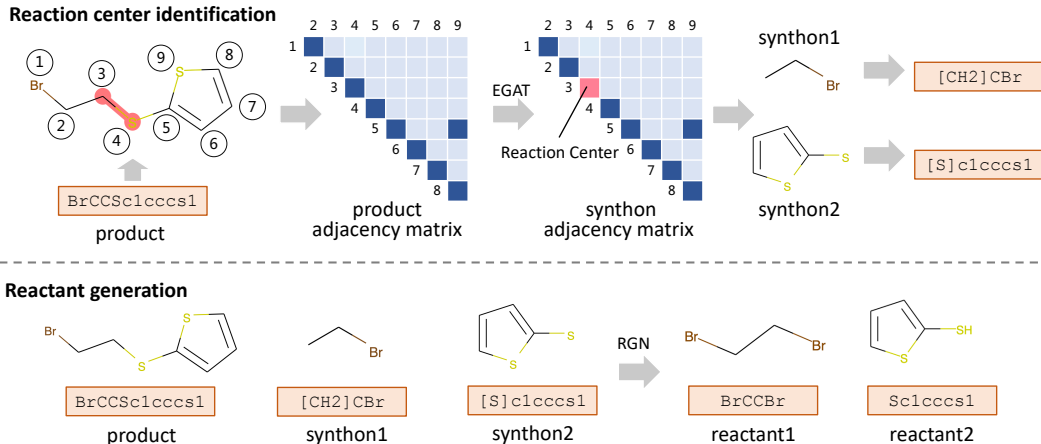

Figure 1: Pipeline overview. We conduct retrosynthesis in two closely dependent steps **reaction center identification** and **reactant generation**. The first step aims to identify the reaction center of the target molecule and generates intermediate synthons accordingly. The second step is to generate the desired set of reactants. Note that a molecule can be represented in two equivalent representations: molecule graph and SMILES string.

denote as $P, S, R$ the product, synthons, and reactants in the reaction formulation, respectively. The single-step retrosynthesis problem can be described as given the desired product $P$, seeking for a set of reactants $R = \{R_1, R_2, ..., R_n\}$ that can produce the major product $P$ through a valid chemical reaction. It is denoted as $P \rightarrow R$ (predict $R$ given $P$), which is the reverse process of the forward reaction prediction problem [20, 21] that predicts the outcome products given a set of reactants.

As illustrated in Figure 1, our method decomposes the retrosynthesis task ($P \rightarrow R$) into two closely dependent steps **reaction center identification** ($P \rightarrow S$) and **reactant generation** ($S \rightarrow R$). The first step is to identify the potential reaction bonds which will be disconnected during the retrosynthesis, and then the product $P$ can be split into a set of intermediate synthons $S = \{S_1, S_2, ..., S_n\}$. Note that each synthon $S_i$ can be regarded as the substructure of a reactant $R_i$. The second step is to transform synthons $S = \{S_1, S_2, ..., S_n\}$ into associated reactants $R = \{R_1, R_2, ..., R_n\}$. Although the intermediate synthons are not needed in retrosynthesis, decomposing the original retrosynthesis task ($P \rightarrow R$) into two dependent procedures can have multiple benefits, which will be elaborated thoroughly in the following sections.

## 2.1 EGAT for reaction center identification

We treat the reaction center identification as a graph-to-graph transformation problem which is similar to the forward reaction outcome prediction [21]. To achieve this, we propose a graph neural network named Edge-enhanced Graph Attention Network (EGAT) which takes the molecule graph $\mathbf{G}$ as input and predicts disconnection probability for each bond, and this is the main task. Since a product may be produced by different reactions, there can be multiple reaction centers for a given product and each reaction center corresponds to a different reaction. While current message passing neural networks [22] are shallow and capture only local structure information for each node, and it is difficult to distinguish multiple reaction centers without global information. To alleviate the problem, we add a graph-level auxiliary task to predict the total number of disconnection bonds.

As shown in Figure 2, distinct from the Graph Attention Network (GAT) [23] which is designed to learn node and graph-level embeddings, our proposed EGAT also learns edge embedding. It identifies the reaction center by predicting the disconnection probability for each bond taking its edge embedding as input. Given the target $\mathbf{G} = \{A, E, X\}$, the EGAT layer computes node embedding $h_i'$ and edge embedding $p_{i,j}'$ from previous layer's embeddings $h_i$ and $p_{i,j}$ by following equations:

where $\mathbf{W} \in \mathbb{R}^{F' \times F}$ , $\mathbf{a} \in \mathbb{R}^{2F'+D}$ , $\mathbf{U} \in \mathbb{R}^{F \times (F'+D)}$ , and $\mathbf{V} \in \mathbb{R}^{D \times (2F+D)}$ are trainable parameters, $||$ means concatenation operation, $\mathcal{N}_i$ is all neighbor nodes of the node $i$, $\alpha_{i,j}$ is the attention weight between the node $i$ and its neighbor node $j$, and $h_i' \in \mathbb{R}^F$ as well as $p_{i,j}' \in \mathbb{R}^D$ are

$$z_i = \mathbf{W}h_i,$$

$$c_{i,j} = \text{LeakyReLU}(\mathbf{a}^T[z_i||z_j||p_{i,j}]),$$

$$\alpha_{i,j} = \frac{\exp(c_{i,j})}{\sum_{k \in \mathcal{N}_i} \exp(c_{i,k})},$$

$$h_i' = \sigma\big(\sum_{j \in \mathcal{N}_i} \alpha_{i,j} \mathbf{U}[z_j||p_{i,j}]\big),$$

$$p_{i,j}' = \mathbf{V}[h_i'||h_j'||p_{i,j}], \tag{1}$$

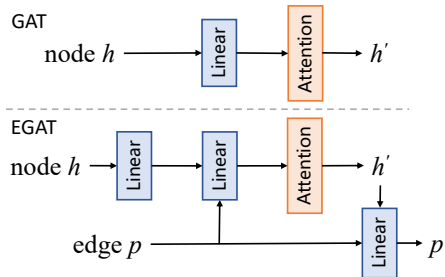

Figure 2: Embedding computation flows of GAT and the proposed EGAT.

the output node and edge representations, respectively. Initial input embeddings $h_i, p_{i,j}$ are the input node and edge feature vectors $x_i, e_{i,j}$, respectively, which will be detailed later, and in this special case the dimensions $F$ and $D$ equals to the dimensions of associated features, respectively.

After stacking multiple EGAT layers, we obtain the final edge representation $p_{i,j}$ for the chemical bond between nodes $i$ and $j$, as well as the node representation $h_i$ for each node $i$. To predict the disconnection probability for a bond, we perform a fully-connected layer parameterized by $\mathbf{w}_{fc} \in \mathbb{R}^D$ and a $Sigmoid$ activation layer to $p_{i,j}$ and its disconnection probability is $d_{i,j} = \text{Sigmoid}(\mathbf{w}_{fc}^T \cdot p_{i,j})$. Note that the multi-head attention mechanism can also be applied like the original GAT. The optimization goal for bond disconnection prediction is to minimize the negative log-likelihood between prediction $d_{i,j}$ and ground-truth $y_{i,j} \in \{0,1\}$ through the binary cross entropy loss function:

$$\mathcal{L}_{\text{M}} = -\frac{1}{K} \sum_{k=1}^{K} \sum_{a_{i,j} \in \mathbf{A}_k} a_{i,j} [(1 - y_{i,j})\log(1 - d_{i,j}) + y_{i,j}\log(d_{i,j})], \tag{2}$$

where $K$ is the total number of training reactions and bond $(i,j)$ exists if the associated adjacency element $a_{i,j}$ is nonzero. The ground truth $y_{i,j} = 1$ means the bond $(i,j)$ is disconnected otherwise remaining the same during the reaction. Bond disconnection labels can be obtained by comparing molecule graphs of target and reactants.

The input of the auxiliary task is the graph-level representation $h_G = \text{READOUT}(\{h_i | 1 \le i \le N\})$, which is the output of the READOUT operation over all learned node representations. We adopts an arithmetic mean as the READOUT function $h_G = \frac{1}{N} \sum_{i=1}^{N} h_i$ and it works well in practice.

Similarly, a fully-connected layer parameterized by $\mathbf{W}_s \in \mathbb{R}^{(1+N_{max}) \times F}$ and a $Softmax$ activation function are applied to $h_G$ to predict the total number of disconnected bonds, which is solved as a classification problem here. Each category represents the exact number of disconnected bonds, so there are $1+N_{max}$ classification categories. $N_{max}$ is the maximum number of possible disconnected bonds in the retrosynthesis. We denote the $Softmax$ output as $q = \text{Softmax}(\mathbf{W}_s \cdot h_G)$. The total number of disconnected bonds for each target molecule is predicted as:

$$n^* = \arg\max_n(q_n) = \arg\max_n(\text{Softmax}(\mathbf{W}_s \cdot h_G)_n), 0 \le n \le N_{max}. \tag{3}$$

The ground truth number of disconnections for molecule $k$ is denoted as $N_k$, the indicator function $\mathbb{1}(i, N_k)$ is 1 if $i$ equals to $N_k$ otherwise it is 0, and the cross entropy loss for the auxiliary task:

$$\mathcal{L}_{\text{A}} = \frac{1}{K} \sum_{k=1}^{K} \text{CrossEntropy}(N_k, q^k) = -\frac{1}{K} \sum_{k=1}^{K} \sum_{i=0}^{N_{max}} \mathbb{1}(i, N_k)\log(q_i^k). \tag{4}$$

Finally, the overall loss function for the EGAT is $\mathcal{L}_{\text{EGAT}} = \mathcal{L}_{\text{M}} + \alpha\mathcal{L}_{\text{A}}$, where $\alpha$ is fixed to 1 in our study since we empirically find that $\alpha$ is not a sensitive hype-parameter.

**Atom and bond features.** The atom feature consists of a series of general atom information such as atom type, hybridization, and formal charge, while the bond feature is composed of chemical bond

information like bond type and conjugation (see Appendix B for details). These features are similar to those used in [24] which is for chemical property prediction. We compute these features using the open-source toolkit RDKit [4]. To fully utilize the provided rich atom-mapping information of the USPTO datasets [19] [25], we add a semi-templates indicator to atom feature. For retrosynthesis dataset with given reaction type, a type indicator is also added to the atom feature.

**Semi-templates.**    For atom-mapped USPTO datasets, reaction templates are extracted from reaction data like previous template-based methods [12, 14, 5]. However, we are not interested in full reaction templates since these templates are often too specific. There are as many as 11,647 templates for the USPTO-50K train data [5]. Only the product side of templates are kept instead, which we name as semi-templates. Since reaction templates are closely related to the exact reaction, the semi-templates indicator expected to play a significant role in reaction center identification.

The semi-templates can be considered as subgraph patterns within molecules. We build a database of semi-templates from training data and find all appeared semi-templates within each molecule. For each atom, we mark the indicator bits associated with appeared semi-templates. Note that each atom within a molecule may belong to several semi-templates since these semi-templates are not mutually exclusive. Although reaction templates are introduced, our method is still template-free since i) only semi-templates are incorporated and our method does not rely on full templates to plan the retrosynthesis, and ii) our EGAT still works well in the absence of semi-templates, with only slight performance degradation (Appendix D.2).

## 2.2    Reactant generation network

Once the reaction center has been identified, synthons can be obtained by applying bond disconnection to decompose the target graph. Since each synthon is basically a substructure within the reactant, we are informed of the total number of reactants and substructures of these reactants. The remaining task $S \rightarrow R$ is much simpler than the original $P \rightarrow R$ in which even the number of reactants is unknown.

Specifically, task $S \rightarrow R$ is to generate the set of desired reactants given obtained synthons. Based on commonsense knowledge of chemical reaction, we propose that the ideal RGN should meet following three requirements: R1) be permutation invariant and generate the same set of reactants no matter the order of synthons, R2) all given information should be considered when generating any reactant, and R3) the generation of each reactant also depends on those previously generated reactants.

To fulfill these requirements, we represent molecules in SMILES and formulate $S \rightarrow R$ as a sequence-to-sequence prediction problem. We convert synthon graphs to SMILES representations using RDKit, though these synthons may be chemically invalid. As in Figure 3, source sequence is the concatenation of possible reaction types, canonical SMILES of the product, and associated synthons. The target sequence is the desired reactants arranged according to synthons.

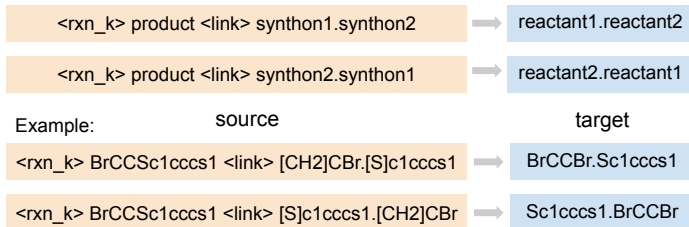

Figure 3: Illustration of source and target sequences. <rxn_k> is the $k$th reaction type if applicable. The product and synthons are separated with a special <link> token. The order of reactants is arranged according to synthons. SMILES strings are joined with a dot following RDkit.

We approximate the requirement R1 by augmenting train samples with reversely arranged synthons and reactants as shown in Figure 3. Our empirical studies demonstrate that such approximation works pretty well in practice. To satisfy the requirement R2, the encoder-decoder attention mechanism [26] [27] is employed, which allows each position in the target sequence attends to all positions in the source sequence. A similar masked self-attention mechanism [27], which masks future positions in the decoder, is adopted to make the RGN meet the requirement R3.

Motivated by the success of Transformer [27] in natural machine translation, we build the RGN based on the Transformer module. Transformer is a sequence-to-sequence model equipped with two types of attention mechanisms: self-attention and encoder-decoder attention [27]. Transformer is also adapted for reaction outcome prediction [28] and retrosynthesis [6], in which both products and reactants are represented in SMILES. We include a brief description of Transformer in Appendix C.

**Determine the generation order of reactants.**   For the first time, the generation order of reactants can be determined by aligning reactants in the target with synthons in the source, thanks to intermediate synthons which are associated with reactants uniquely. While the generation order of reactants is undetermined in previous methods [4, 6, 15], which naively treats the sequence-to-sequence model as a black box. The uncertainty of the generation order makes their models hard to train.

**Robustify the RGN.**   We find the EGAT suffers from distinguishing multiple coexisting reaction centers, which is the major bottleneck of our method. As a result of the failure of identifying the reaction center, the generated synthons are different from the ground truth. To make our RGN robust enough and able to predict the desired reactants even if the EGAT fails to recognize the reaction center, we further augment RGN training data by including those unsuccessfully predicted synthons on training data. We do not reverse the order of synthons for these augmentation samples like in Figure 3. The intuition behind is that EGAT tends to make similar mistakes on training and test datasets since both datasets follow the same distribution. This method can make our RGN able to correct reaction center prediction error and generate the desired set of reactants.

## 3   Experiments

**Dataset and preprocessing.**   We evaluate our method on USPTO-50K [19] and USPTO-full [25] to verify its effectiveness and scalability. USPTO-50K consists of 50K reactions annotated with 10 reaction types (see appendix A for type distribution), which is derived from USPTO granted patents [29]. It is widely used in previous retrosynthesis work. We adopt the same training/validation/test splits in 8:1:1 as [12, 5]. For RGN training data, we add an extra 28K samples of which synthons are reversed as shown in Figure 3 if there are at least two synthons. There are 68K training samples for RGN, which is still denoted as USPTO-50K in the following content. The USPTO-full consists of 950K cleaned reactions from the USPTO 1976-2016 [25], which has 1,808,937 raw reactions without reaction types. Reactions with multiple products are duplicated into multiple single-product ones. After removing invalid reactions (empty reactant and missing atom mappings) and deduplication, we can obtain 950K reactions [5], which are randomly partitioned into training/validation/test sets in 8:1:1.

For the EGAT, we build molecule graphs using DGL [30] and extract atom and bond features with RDkit. By comparing molecule graphs of product and reactants, we can identify disconnection bonds within the product graph and obtain training labels for both main and auxiliary tasks. This comparison can be easily done for atom-mapped reactions. For reactions without atom-mapping, a substructure matching algorithm in RDKit can be utilized to accomplish the comparison. We use RDChiral [31] to extract super general reaction templates, and obtain 1859 semi-templates for USPTO-50K training data. Semi-templates that appear less than twice are filtered and finally 654 semi-templates are obtained. As for the RGN, the product molecule graph is divided into synthon graphs according to the ground truth reaction center, then are converted into SMILES strings. The input sequence of RGN is the concatenation of the possible reaction type, product SMILES string, and synthon SMILES strings as illustrated in Figure 3.

**Implementation.**   All reactions are represented in canonical SMILES, which are tokenized with the regular expression in [32]. We use DGL [30] and OpenNMT [33] to implement our EGAT and RGN models, respectively. As for the EGAT, we stack three identical four-head attentive layers of which the hidden dimension is 128. All embedding sizes in EGAT are set to 128, such as $F$, $F'$, and $D$. The $N_{max}$ is set to be two to cover 99.97% training samples. We train the EGAT on USPTO-50K for 80 epochs. EGAT parameters are optimized with Adam [34] with default settings, and the initial learning rate is 0.0005 and it is scheduled to multiply 0.2 every 20 epochs. We train the RGN for $300,000$ time steps, and it takes about 30 hours on two GTX 1080 Ti GPUs. We save a checkpoint of

RGN parameters every $10,000$ steps and average the last 10 checkpoints as the final model. We run all experiments for three times and report the means of their performance in default.

**Evaluation metric.** The Top-$N$ accuracy is used as the evaluation metric for retrosynthesis. Beam search [35] strategy is adopted to keep top K predictions throughout the reactant generation process. K is set to 50 in all experiments. The generated reactants are represented in canonical SMILES. A correct predicted set of reactants must be exactly the same as the ground truth reactants.

## 3.1 Reaction center identification results

To verify the effectiveness of edge-enhanced attention mechanism, we also include the ablation study by removing edge embedding $p_{i,j}$ when computing the coefficient $c_{i,j} = \text{LeakyReLU}(\mathbf{a}^T[z_i||z_j])$. Results are reported in Table 1. The auxiliary task (**Aux**) can successfully predict the number of disconnection bonds for 99.2% test molecules given the reaction type (**Type**) while 86.4% if not given. As for the main task (**Main**) alone, its prediction accuracy is 74.4% w/ reaction type and 51.5% wo/ reaction type. However, if we adopt the prediction from the auxiliary task as the prior of the number of disconnection bonds, and select the most probable disconnection bonds (**EGAT**), then the prediction accuracy can be boosted to 86.0% (w/) and 64.9% (wo/), respectively. The edge-enhanced attention (**EAtt**) can consistently improve the model's performance in all tasks. The improvement is more significant when the reaction type is unknown, so our EGAT is more practical in real world applications without reaction types. This demonstrates that the reaction type information plays an important role in the retrosynthesis. The reactions of the same type usually share similar reaction patterns (involved atoms, bonds, and functional groups), it is much easier to recognize the reaction center if the reaction type is given as the prior knowledge. We also verify the importance of semi-templates in Appendix D.2.

Table 1: Results of EGAT on USPTO-50K dataset. *EAtt* and *Aux* are the short for edge-enhanced attention and auxiliary task, respectively. *EGAT* consists of both main and auxiliary tasks. The prediction is binarized with a threshold of 0.5 if main task alone.

| Type | EAtt | Accuracy (%) | | |
|:---:|:---:|:---:|:---:|:---:|
| | | Main | Aux | EGAT |
| ✓ | ✗ | 73.9 | 99.1 | 85.7 |
| ✓ | ✓ | **74.4** | **99.2** | **86.0** |
| ✗ | ✗ | 50.0 | 86.1 | 64.3 |
| ✗ | ✓ | **51.5** | **86.4** | **64.9** |

## 3.2 Reactant prediction results

To robustify the RGN as described in the paragraph **Robustify the RGN**, we also conduct the $P \rightarrow S$ prediction on the EGAT training data for USPTO-50K (40K), and the prediction accuracy is 89.0% for the reaction type conditional setting. We can obtain about 4K unsuccessful synthon predictions as augmentation samples (**Aug**), adding the original 68K RGN training data, the total RGN training data size is 72K. For the unconditional setting, the EGAT accuracy is 70.0% and there are 12K augmentation samples, and the total RGN training size is 80K in this case. We train RGN models on the USPTO-50K with/without the augmentation (**Aug**), and report results in Table 2.

**RGN evaluation** For the RGN evaluation, the RGN input consists of the ground truth synthons. Therefore the results in Table 2 indicate the upper bound of our method's overall retrosynthesis performance. The proposed augmentation strategy does not always improve the upper bound. Without given reaction type, the RGN generally performs worse with the augmentation due to the introduced dirty training samples. However, when given reaction type, this augmentation boosts its prediction accuracy. We presume that it is because the reaction type plays a significant role. The RGN learns to put more attention on the reaction type and product instead of synthons to generate the reactants.

**Retrosynthesis evaluation** To evaluate the overall retrosynthesis prediction accuracy, the generated synthons from $P \rightarrow S$ instead of the ground truth are input into the RGN. In this way, we only need to compare the predicted reactants with the ground truth ones, without considering if the reaction center predictions correct or not. We report the retrosynthesis results in Tables 3. Our method RetroXpert achieves impressive performance on the test data. Specifically, when given reaction types, our proposed method achieves 70.4% Top-1 accuracy, which outperforms the SOTA Top-1 accuracy 63.2% [5] by a large margin. Note that our Top-1 accuracy 70.4% is quite close to the upper bound

Table 2: $S \rightarrow R$ prediction results. *Aug* denotes training data augmentation. Evaluation results are based on ground-truth synthons as the RGN input.

| Type | Aug | Training size | Top-$n$ accuracy (%) | | | | | |
|:---:|:---:|:---:|:---:|:---:|:---:|:---:|:---:|:---:|
| | | | 1 | 3 | 5 | 10 | 20 | 50 |
| ✓ | ✗ | 68K | 72.9 | 86.5 | 88.3 | 89.5 | 90.4 | 91.6 |
| ✓ | ✓ | 72K | **73.4** | **86.7** | **88.5** | **89.7** | **90.9** | **92.1** |
| ✗ | ✗ | 68K | **71.9** | **85.7** | **87.5** | **88.9** | **90.0** | **91.0** |
| ✗ | ✓ | 80K | 70.9 | 84.6 | 86.4 | 88.2 | 89.4 | 90.6 |

Table 3: Retrosynthesis results compared with the existing methods. NeuralSym [13] results are copied from [5]. *We run the self-implemented SCROP [6] and official implementation of GLN [5] on the USPTO-full dataset.

| Methods | Top-$n$ accuracy (%) | | | | | |
|:---|:---:|:---:|:---:|:---:|:---:|:---:|
| | 1 | 3 | 5 | 10 | 20 | 50 |
| Reaction types given as prior on USPTO-50K | | | | | | |
| Seq2Seq [4] | 37.4 | 52.4 | 57.0 | 61.7 | 65.9 | 70.7 |
| RetroSim [12] | 52.9 | 73.8 | 81.2 | 88.1 | 91.8 | 92.9 |
| NeuralSym [13] | 55.3 | 76.0 | 81.4 | 85.1 | 86.5 | 86.9 |
| SCROP [6] | 59.0 | 74.8 | 78.1 | 81.1 | - | - |
| GLN [5] | 63.2 | 77.5 | 83.4 | **89.1** | **92.1** | **93.2** |
| RetroXpert | **70.4** | **83.4** | **85.3** | 86.8 | 88.1 | 89.3 |
| Reaction type unknown on USPTO-50K | | | | | | |
| RetroSim [12] | 37.3 | 54.7 | 63.3 | 74.1 | 82.0 | 85.3 |
| NeuralSym [13] | 44.4 | 65.3 | 72.4 | 78.9 | 82.2 | 83.1 |
| SCROP [6] | 43.7 | 60.0 | 65.2 | 68.7 | - | - |
| GLN [5] | 52.6 | 68.0 | 75.1 | 83.1 | **88.5** | **92.1** |
| RetroXpert | **65.6** | **78.7** | **80.8** | **83.3** | 84.6 | 86.0 |
| Retrosynthesis results on USPTO-full. | | | | | | |
| GLN* [5] | 39.0 | 50.1 | 55.3 | 61.3 | 65.9 | 69.1 |
| SCROP* [6] | 45.7 | 60.7 | 65.3 | 70.1 | 73.3 | 76.0 |
| RetroXpert | **49.4** | **63.6** | **67.6** | **71.6** | **74.6** | **77.0** |

73.4% in Table 2, which indicates the proposed augmentation strategy in **Robustify the RGN** is considerably effective. As for results wo/ given reaction type, our model improves the SOTA Top-1 accuracy from 52.6% [5] to 65.6%. To verify the effectiveness of augmentation, we conduct ablation study and report results in Appendix D.3.

While our method outperforms in Top-1, Top-3, and Top-5 accuracy, template-based methods GLN [5] and RetroSim [12] are better at Top-20 and Top-50 predictions since they enumerate multiple different reaction templates for each product to increase the hit rate. While our RetroXpert is currently designed to find the best set of reactants. To increase the diversity, we can design new strategies to enumerate multiple reaction centers for each product. This is left as the feature work.

We notice that the gap between Top-2 and Top-1 accuracy is around 10%. After investigating these 10% predictions by experienced chemists from the synthetic chemistry perspective, we find about 9/10 these Top-1 predictions are actually reasonable (see Appendix E for details). This indicates that our method can learn general chemical reaction knowledge, which is beyond the given ground truth.

## 4  Large scale experiments

To demonstrate the scalability of our method, we also experiment on the USPTO-full dataset, which consists of 760K training data. We extract 75,129 semi-templates and keep only 3,788 ones that appear at least 10 times. We set $N_{max}$ as 5 to cover 99.87% training data. We obtain 1.35M training data after reversing synthons. The final accuracy of the $P \to S$ on training set is 60.5%, and there are 0.3M unsuccessful synthon data and the total RGN training data size is 1.65M. We train the RGN for 500,000 time steps on USPTO-full while keeping the other settings the same as those in section 3. We run the official implementation of GLN following their instructions [5], as well as the self-implemented SCROP [6] on the USPTO-full dataset. Experimental results are reported at the bottom of Table 3. Our method again significantly outperforms the SCROP and GLN, which demonstrates that our model scales well to the large real-world dataset. Note that both template-free methods SCROP and RetroXpert outperform the GLN significantly, which may indicate the scalability of template-based methods is very limited.

## 5  Prediction visualization

For EGAT, how the auxiliary task helps to identify the reaction center is illustrated in Figure 4. Note that in the first example the two colored bonds and their surrounding structures are very similar. Current shallow GNNs consider only local information and fails to distinguish the true reaction center. Under the guidance of the auxiliary task, EGAT is able to identify the true reaction center. Figure 5 demonstrates the robustness of our method. Even if the predicted synthons are different from the ground truth, the RGN still successfully generates desired reactants.

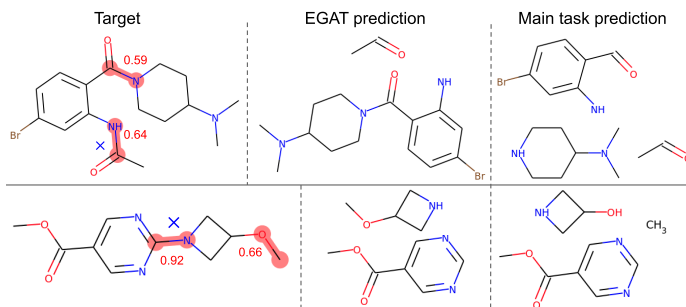

Figure 4: Importance of the auxiliary task. Pink indicates the reaction center along with disconnection probability predicted by the EGAT main task. Blue cross indicates the ground truth disconnection. Our EGAT successfully finds the desired reaction center under the guidance of the auxiliary task.

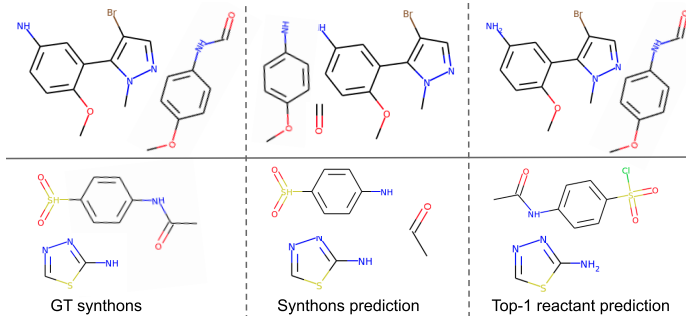

Figure 5: Robustness of the RGN. Top-1 prediction is the same to ground truth.

## 6  Discussion

One major common limitation of current retrosynthesis work is the lack of reasonable evaluation metrics. There may be multiple valid ways to synthesize a product, while the current evaluation metric considers only the given reaction. More evaluation metrics should be proposed in the future.

## Broader Impact

Our proposed new retrosynthesis method RetroXpert solves the retrosynthesis prediction in two steps like chemists do, and it achieves impressive performance. It is template-free and is very scalable to the large real-world dataset. We believe that our work will greatly inspire and advance related research, such as forward reaction prediction and drug discovery. The researchers and industry experts in drug discovery will benefit most from this research since the retrosynthesis prediction is an important part of drug discovery. We are not aware anyone may be put at disadvantage from this research. Our method does not take advantage of the data bias, it is general and scalable.

## Acknowledgments and Disclosure of Funding

We would like to thank Hanjun Dai for providing the source implementation of GLN. This work was partially supported by US National Science Foundation IIS-1718853, the CAREER grant IIS-1553687 and Cancer Prevention and Research Institute of Texas (CPRIT) award (RP190107).

## Footnotes

[3]https://www.daylight.com/dayhtml/doc/theory/theory.smiles.html

[4] https://www.rdkit.org

[5]Code and processed USPTO-full data are available at https://github.com/uta-smile/RetroXpert

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
