[Supplementary Material]

# Appendices

## A    Dataset information

The USPTO-50K dataset is annotated with 10 reaction types, the distribution of reaction types is displayed in Table 4. The distribution is extremely unbalanced. We also report the statistics of the number of disconnection bonds for training reactions in Tables 5 and 6.

Table 4: Distribution of 10 recognized reaction types.

| Reaction type | Reaction type name | # Examples |
|---|---|---|
| 1 | Heteroatom alkylation and arylation | 15204 |
| 2 | Acylation and related processes | 11972 |
| 3 | C-C bond formation | 5667 |
| 4 | Heterocycle formation | 909 |
| 5 | Protections | 672 |
| 6 | Deprotections | 8405 |
| 7 | Reductions | 4642 |
| 8 | Oxidations | 822 |
| 9 | Functional group interconversion (FGI) | 1858 |
| 10 | Functional group addition (FGA) | 231 |

Table 5: Statistics of the number of disconnection bonds for the USPTO-50K training reactions.

| # Disconnection bonds | 0 | 1 | 2 | $\geq 3$ |
|---|---|---|---|---|
| # Reactions | 11296 | 27851 | 849 | 12 |
| Accumulative percent | 28.23% | 97.85% | 99.97% | 100.00% |

Table 6: Statistics of the number of disconnection bonds for the USPTO-full training reactions.

| # Disconnection bonds | 0 | 1 | 2 | 3 | 4 | 5 | $\geq 6$ |
|---|---|---|---|---|---|---|---|
| # Reactions | 161500 | 485449 | 88146 | 19303 | 5687 | 2032 | 1000 |
| Accumulative percent | 21.16% | 84.77% | 96.33% | 98.86% | 99.60% | 99.87% | 100.00% |

## B    Atom and bond features

Table 7: Atom Features used in EGAT. All features are one-hot encoding, except the atomic mass is a real number scaled to be on the same order of magnitude. The upper part is general atom feature following [24], the lower part is specifically extended for the retrosynthesis prediction. Semi-templates size is 654 for the USPTO-50K dataset.

| Feature | Description | Size |
|---|---|---|
| Atom type | Type of atom (ex. C, N, O), by atomic number. | 100 |
| # Bonds | Number of bonds the atom is involved in. | 6 |
| Formal charge | Integer electronic charge assigned to atom. | 5 |
| Chirality | Unspecified, tetrahedral CW/CCW, or other. | 4 |
| # Hs | Number of bonded Hydrogen atom. | 5 |
| Hybridization | sp, sp2, sp3, sp3d, or sp3d2. | 5 |
| Aromaticity | Whether this atom is part of an aromatic system. | 1 |
| Atomic mass | Mass of the atom, divided by 100. | 1 |
| Semi-templates | Semi-templates that the atom is within. | 654 |
| Reaction type | The specified reaction type if it exists. | 10 |

Table 8: Bond features used in EGAT. All features are one-hot encoding.

| Feature | Description | Size |
|---|---|---|
| Bond type | Single, double, triple, or aromatic. | 4 |
| Conjugation | Whether the bond is conjugated. | 1 |
| In ring | Whether the bond is part of a ring. | 1 |
| Stereo | None, any, E/Z or cis/trans. | 6 |

## C  Transformer

The transformer [27] is an autoregressive encoder-decoder model built with multi-head attention layers and position-wise feed-forward layers. As illustrated in Figure 6, the encoder is composed of stacked multi-head self-attention layers and position-wise feed-forward layers. The encoder self-attention layers attend the full input sequence and iteratively transform it into a latent representation with the self-attention mechanism. The decoder is similar to the encoder. In addition to multi-head self-attention layers and position-wise feed-forward layers, the multi-head encoder-decoder attention layers are inserted to perform cross attention over the encoder output. Different from the encoder self-attention layers, the decoder adopts the masked self-attention which prevents the decoder positions from attending future positions. The encoder-decoder attention and masked self-attention layers enable the decoder to combine the information from the source sequence and the target sequence that has been produced to make the output prediction. We refer readers to [27] and The Illustrated Transformer for comprehensive details about the Transformer.

Figure 6: Transformer model architecture. The residual connection and layer normalization layer are omitted in the illustration for simplification.

The transformer removes all recurrent units and introduces a positional encoding to account for the order information of the sequence. Positional encoding adds a position-dependent signal to the token embedding of size $d_{emb}$ to discriminate the position of different tokens in the sequence:

$$PE_{(pos,2i)} = \sin \frac{pos}{10000^{2i/d_{emb}}}, PE_{(pos,2i+1)} = \cos \frac{pos}{10000^{2i/d_{emb}}} \tag{5}$$

where $pos$ is the token position and $i$ is the dimension of the positional encoding.

The transformer adopts a scale dot-product attention as the attention formulation, which compute the attention weighted output by taking as input the matrix represented keys K, values V, and queries Q:

$$\text{Attention}(Q, K, V) = \text{Softmax}(\frac{QK^T}{\sqrt{d_k}})V \tag{6}$$

where the $d_k$ is the dimension of Q and K.

## C.1 Parameters setting

We compose both the encoder and decoder of four layers of size 256. The label smoothing parameter is set to 0 since a nonzero label smoothing parameter will deteriorate the model's discrimination [28]. We adopt eight attention heads as suggested. We set the batch size to 4096 tokens and accumulate gradients over four batches.

# D More experimental results

## D.1 Per category performance

Figure 7: Top-10 retrosynthesis accuracy per reaction category with given reaction type.

Figure 8: Top-10 retrosynthesis accuracy per reaction category with unknown reaction type.

We investigate the retrosynthesis accuracy per reaction category on USPTO-50K to have a better understanding of our model's capability in different types of reaction. We report Top-10 accuracy for a fair comparison following baseline methods, though our method is not designed for diverse predictions. Although the reaction types are highly imbalanced as shown in Table 4, our method

performs well in all reaction categories as displayed in Figure 7, which indicates our method is not that sensitive to the number of samples of the same reaction type. Particularly, our method achieves comparable or better performance in 9 out 10 reaction categories compared with the template-based methods RetroSim and GLN. For rare categories like class 4 and 10, our methods is much better than the GLN. Similar conclusion also applies to the unknown reaction type scenario as illustrated in Figure 8. Note that for the most common type 1 and 2, the type prior does not help much to our method's performance, which suggests that our method may exploit training reactions to the maximum extent given enough reaction data.

## D.2 Ablation study of atom features

Our method can also work without semi-templates. When removing semi-templates, the EGAT performance drops slightly as listed in Table 9. The semi-templates feature is not a must component of our method, but it is definitely helpful for finding the reaction center.

Table 9: Results of atom features ablation study. *Aux* is the short for auxiliary. *EGAT* consists of both main and auxiliary tasks. The prediction is binarized with a threshold of 0.5 if the main task alone.

| Type | Semi-templates | Accuracy (%) | | |
|:---:|:---:|:---:|:---:|:---:|
| | | Main | Aux | EGAT |
| ✓ | ✗ | 70.0 | **99.2** | 84.0 |
| ✓ | ✓ | **74.4** | **99.2** | **86.0** |
| ✗ | ✗ | 43.3 | 83.8 | 59.9 |
| ✗ | ✓ | **51.5** | **86.4** | **64.9** |

## D.3 Ablation study of augmentation in RGN

Table 10: Retrosynthesis results of augmentation ablation study.

| Training Aug | Test Aug | Top-$n$ accuracy (%) | | | | | |
|:---:|:---:|:---:|:---:|:---:|:---:|:---:|:---:|
| | | 1 | 3 | 5 | 10 | 20 | 50 |
| Reaction type given as prior on USPTO-50K | | | | | | | |
| ✗ | ✗ | 63.5 | 75.2 | 76.7 | 77.6 | 78.3 | 79.3 |
| ✗ | ✓ | 64.0 | 75.7 | 77.5 | 78.3 | 79.2 | 80.2 |
| ✓ | ✗ | 63.8 | 75.1 | 76.5 | 77.4 | 78.4 | 79.3 |
| ✓ | ✓ | **70.4** | **83.4** | **85.3** | **86.8** | **88.1** | **89.3** |
| Reaction type unknown on USPTO-50K | | | | | | | |
| ✗ | ✗ | 48.1 | 56.3 | 57.3 | 58.0 | 58.7 | 59.1 |
| ✗ | ✓ | 48.4 | 56.9 | 57.9 | 58.8 | 59.6 | 60.2 |
| ✓ | ✗ | 47.6 | 56.1 | 57.0 | 57.9 | 58.5 | 59.2 |
| ✓ | ✓ | **65.6** | **78.7** | **80.8** | **83.3** | **84.6** | **86.0** |

We robustify the RGN by including unsuccessfully predicted synthons by EGAT as the RGN training data augmentation (**Training Aug**). When evaluating the retrosynthesis on test data, we also gather predicted synthons from the EGAT to form the RGN input sequences without considering if the reaction center identification successful or not, and we denote this evaluation strategy as the test augmentation (**Test Aug**) with a little abused use of augmentation. Without the test augmentation, we must first evaluate reaction center identification results, and only try to predict reactants for the product whose reaction center is successfully identified. In this case, the overall retrosynthesis performance will be capped by the EGAT. The ablation study results are listed in Table 10.

Generally, applying only training or test augmentation makes only tiny influence on the retrosynthesis performance in both cases (w/ or wo/ reaction type). While the retrosynthesis performance will be boosted significantly if both training and test augmentation are adopted. This is not unexpected. Exploiting training data augmentation makes the RGN robust and also assigns a correction ability to the RGN. The correction ability will take effect only if the test augmentation is also employed.

# E  Top-1 and Top-2 predictions

About 10% Top-1 predictions by our model have been considered as wrong predictions while the associated Top-2 predictions are the same to the ground-truth. However, 9 in 10 of these Top-1 predictions are re-considered as reasonable and valid predictions checked by experienced chemists from the synthetic chemistry perspective. As Figure 9 shows, the major retro-predictions that both Top-1 and Top-2 can be thought correct, are among metal-catalyzed cross-coupling reactions, N- and O-alkylation reactions, saponification of ethyl esters and methyl esters, different sources of reactants, esterification of alcohol with acyl chlorides or carboxylic acid, and deprotection of different protecting groups to same alcohols.

There are some deprotection reactions with different protecting groups, such as deprotecting O-THP ether and O-Bn ether to free alcohol in Figure 9(a). They are prevalent strategies in chemistry utilizing different protecting groups. In Figure 9(b), both bromoarenes and iodoarenes are reactive enough to initiate Suzuki coupling reactions, similar to N- and O-alkylation of propargyl like or benzyl chloride and bromide in Figure 9(c). In Figure 9(d), hydrolysis of ethyl ester and methyl ester to corresponding carboxylic acid can both occur under certain conditions, although saponification of methyl ester is faster than ethyl ester. Real reactants that participated in the reactions are predicted in our Top-1 predictions, such as allyl Grignard reagent and acyl chloride in cases shown in Figure 9(e). Last but not least, in Figure 9(f), methyl boronic acid or its trimer form and trimethyl borate are very common reagents used by chemists in Suzuki coupling reaction to introduce methyl group.

Figure 9: Top-1 and Top-2 predictions are both reasonable reactants.