[Reviews · NeurIPS 2020]

Review 1

Summary and Contributions: This submission proposes a two-stage approach to one-step retrosynthesis prediction where the overall task of proposing reactants from a product molecule is divided into (1) reaction center identification and synthon generation; (2) reactant completion from synthons.

Strengths: Strong empirical results are obtained on the USPTO-50K benchmark dataset. The approach is novel (note: contemporary work https://arxiv.org/abs/2006.07038) and informed by how domain experts approach the problem. Allowing the Transformer model to correct mistakes from the graph-based model is an interesting and successful approach.

Weaknesses: The main weakness of this work is in the presentation and language. It would be difficult for a non-expert to follow details about the methods at times. One could argue that the architectures used are not novel from a machine learning perspective, but I believe it is an interesting application nonetheless.

Correctness: To ensure the comparison to other approaches is as fair as possible, it will be important to note - How hyperparameters were selected - How the validation dataset was used (early stopping is not mentioned) - Whether the evaluation in Table 2 used the ground truth synthons as input when testing S->R performance.

Clarity: The language throughout the paper could be improved (set notation used when referring to sets, poor grammar in many sentences, spelling out “ground truth” instead of GT). The work also does not have a real discussion section or conclusion, likely due to length restrictions. I have some pickier comments about the contextualization of the work: - L28: Only one of the previous methods the authors cite used hand-coded reaction templates; the others algorithmically extract them. The authors could revise the discussion of template-based methods to more clearly separate inherent limitations in template-based methods from limitations due to manual encoding (L34-35). - L38: Could the authors clarify the clause “which seem to get stuck in an infinite loop”? - L110: When K>1, does that mean that multiple reaction examples with the same product have been merged into one training instance? The use of “total number of reactions” is a little confusing because this is one-step retrosynthesis where there is, by definition, one reaction.

Relation to Prior Work: The distinction from prior work would be clear enough, provided that some minor corrections and clarifications are made (see my other comments). Minor note: the capitalization in many references is messed up.

Reproducibility: No

Additional Feedback: Sources of confusion: - L36: The authors mention that relying on atom mapping is a negative factor, but my impression is that their synthon approach (and template-ish featurization) relies on atom mapping. Please clarify. - The authors talk about MPNNs only capturing local information, but the EGAT network only passes messages between neighboring atoms (i.e., it does not overcome this limitation). - L133 (Semi-templates): Were product templates for featurization extracted from the entire USPTO-50K dataset or just the training set? Because there is some lack of clarity in the methods, I believe it would be important for the authors to make their code available. At the very least, the authors need to make the 950k reactions from the USPTO available for others to benchmark against. ==== I've increased my score from 6 to 7 based on the authors' response


Review 2

Summary and Contributions: This paper presents a novel, template free algorithm for retrosynthesis by automating the protocol typically used by chemists to predict possible reactants when given the product. This involves two steps, (i) identify the potential reaction center and split the molecule according to this hypothesis, and (ii) predict the corresponding reactants. For reaction center identification, the paper presents an interesting Edge-enhanced Graph Attention Network that adds a graph-level auxiliary task to predict the total number of disconnection bonds. The authors then use a transformer to build the reactant generation network. To test their approach, the authors carry out experiments using the well known USPTO-50k and USPTO-full datasets. Performance is evaluated using Top-N accuracy, and the method proposed in this paper outperforms the existing SOTA Top-1 accuracy by some margin on both the USPTO-50k and USPTO-full dataset.

Strengths: This paper presents a novel approach that generates considerable improvement in top-1 performance on the USPTO-50k dataset.

Weaknesses: No comment is made about possible biases in the USPTO dataset, and how these might affect the reported results. It would be good to check whether the method suggested here exploits any of these biases by comparing performance on a different independently constructed dataset. Please could the authors address the question of how their approach to the single step problem can be extended to apply to multistep retrosynthesis challenges. Furthermore, a more comprehensive survey of recent literature in this area should be presented, particularly recent template free approaches.

Correctness: I did not find any errors with the empirical methodology of the paper.

Clarity: There are some minor issues with the language writing throughout, and occasionally these obscure the sense of the paper.

Relation to Prior Work: Please could the authors specifically address whether the general strategy of splitting one-step retrosynthesis into a reaction center prediction step and synthon -> reactant conversion has been attempted before in the literature? For example, I do not find a reference to https://arxiv.org/pdf/2003.12725.pdf which is quite relevant. In terms of overall results, among others I would expect to see a reference to https://arxiv.org/abs/1906.02308. Please could the authors provide a more thorough description of recent work in this space.

Reproducibility: Yes

Additional Feedback:


Review 3

Summary and Contributions: The paper proposes a sensible two stage model to predict retrosynthetic disconnections. First, it predicts which bonds need to be broken to obtain what is called synthons, then it completes the synthons by added the required functional groups. Performance is measured on the standard dataset, and very good performance is achieved.

Strengths: + good description + thorough evaluation, with extra experiments in appendix! + good results + well motivated model + novelty in the way the problem is approached + relevance to neurips community, because tasks that operate on graphs and change their structure can be found in many domains, not just chemistry

Weaknesses: - the title. I would suggest to be more modest. For example, writing "automating the procedure that chemists used to do" in the abstract sounds like automated retrosynthesis is now a solved problem and chemists are not needed anymore. This is very far away in the future.

Correctness: The claim that the model performs retrosynthesis like a chemist is not fully accurate, because chemists use many more complex reasoning processes. It is like saying AlexNet performs image recognition like a human. I would suggest to use something more modest like "inspired from how chemists are taught to approach retrosynthesis ...". These kind of claims distract from the great modelling work that the authors have done. - how do the authors obtain the label for the bond that is being broken? it seems that this comes from the reaction mapping, even though the authors previously claim that relying on the atom mapping is a deficient. so in some sense, the presented model is not fully template free, right? (by the way, I don't think it matters whether a model is template free or not. what matters is good performance).

Clarity: Can the authors maybe elaborate further on the semi-templates indicator? Let me paraphrase my understanding, and the authors can correct me: If I understand correctly, these semi-template subgraphs are essentially the strategic bonds in the target molecule. a list of these subgraphs gets extracted from the training data. now, when making a prediction for a target T the whole list of subgraphs is matched in T, which labels the atoms as a semi-template. is that right?

Relation to Prior Work: In the beginning (before references 3,4,5, I would recommend that the authors cite Segler et al https://doi.org/10.1002/chem.201605499 (neuralsym) which was the first paper to propose machine learning for retrosynthesis, and show that it is actually possible to perform fully ML & data driven retrosynthesis. Furthermore, I would suggest to mention Shi et al. https://arxiv.org/abs/2003.12725 which uses a related idea (but not compare, because it was concurrent work). Also, the concept of predicting bond changes in the forward reaction prediction problem was explored by Bradshaw et al which would be good to cite https://arxiv.org/abs/1805.10970 Also, it is recommended to cite https://arxiv.org/abs/1701.01329 which was the first paper to introduce neural language models (at the time LSTM) to chemistry via SMILES and use them for molecule generation.

Reproducibility: Yes

Additional Feedback: I think this is a good paper, and I will raise my score when the authors are able to address the mentioned issues. _______ post rebuttal: I want to thank the authors for the replies, and I think the paper should be accepted (score=8)

[Author Response · NeurIPS 2020]

We thank all reviewers for their insightful comments and suggestions, which will be incorporated into the revised version. Our source code and the processed 950K UPSTO-full dataset will be released to promote the related future research if accepted. We first address a common concern about the concurrent work G2Gs[1].

The concurrent work G2Gs presents a similar two-step framework, while our method is more general and scalable. G2Gs predicts at most one bond disconnection while our method can predict multiple bond disconnections, which is more general. The scalability of our method is verified on USPTO-full while G2Gs does not scale to USPTO-full since it can only cover 84.77% training reactions (Appendix Table 6). Besides, we formulate the reactant generation as a seq2seq task by representing the molecule/synthon in SMILES, while G2Gs adopts graph representation and generates reactants with the graph generation algorithm. Last but not least, our model can consider already generated reactants when generating the next reactant (see L156 and L172 in our submission), while G2Gs independently generates multiple reactants for each target, which goes against the nature of chemical reaction.

We keep muted about G2Gs before its conference version is available since we have some concerns about it. The major concern is that the detailed composition of atom features is not given in their paper [1]. The minor concern is that they failed to demonstrate the scalability on the USPTO-full. This discussion will be included into our revised version.

**Reviewer 1 Q1: How hyperparameters were selected, and how the validation dataset was used:** We find the optimal hyperparameters with the best performance on the validation dataset. The model is robust to hyperparameters, and we did not exhaustively search for hyperparameters. Both EGAT and RGN are trained for a fixed number of epochs (L222-227), and results of the final EGAT model are reported and early stopping is unnecessary.

**Q2: EGAT network does not overcome the limitation of capturing only local information:** Capturing only local information is a common limitation for MPNNs, and the EGAT belongs to MPNNs. We are not to design a new architecture to overcome the limitation, but to mitigate it with a graph-level task predicting the number of disconnections.

**Q3: The synthon approach and semi-template rely on atom mapping:** Atom mapping is optional for our method. The synthon approach can also work for reactions without provided atom mapping (L208-212). The EGAT performance degrades only slightly when removing the semi-template indicator (L143-146).

**Q4: Could the authors clarify the clause "which seem to get stuck in an infinite loop" (L38):** Manual encoding is infeasible for large reaction datasets. While template extraction algorithms rely on accurate atom mapping, which requires expert rules. It comes back to the infeasibility issue. We will revise the description and language.

**Other questions:** The ground truth synthons are used for the evaluation in Table 2 (L264). L110: K > 1 does not indicate multiple reactions with the same product are merged. K is the total number of reactions in training set, since loss should be aggregated across the full training set. Semi-templates are extracted from training set as indicated at L136. We will revise the discussion of template-based methods (L28, L34-35) as suggested to make it more clear.

**Reviewer 2 Q1: Compare performance on a different independently constructed dataset.** As far as we are concerned, we are not aware of other large and public chemical reaction datasets except for USPTO-50K and USPTO-full.

**Q2: Whether the general strategy of splitting and conversion has been attempted:** This general strategy can be found in the very early work [2]. We instantiate the strategy with deep learning models, which are novel and effective.

**Q3: Multi-step retrosynthesis and related prior work:** Monte Carlo tree search can be applied to the single-step retrosynthesis recursively until reaching available molecules. We will include the suggested related work. Recent template-free approaches formulate the retrosynthesis as a translation problem (L39-42), except for the G2Gs [1].

**Reviewer 3 Q1: How to obtain the label for the bond disconnection:** The label comes from the atom mapping which it is provided by the USPTO datasets. Our method also works for datasets without atom mapping, since the disconnection label can also be obtained using Maximum Common Substructure algorithm (L208-212).

**Q2: Semi-templates indicator:** Sorry for the ambiguity. Your understanding is right. Note that semi-template subgraphs may contain strategic bonds as well as some neighboring bonds and atoms which provide important reaction context. The detailed composition of semi-templates depends on the adopted reaction template extraction algorithm.

**Other questions:** We will make our statement more humble and practical as suggested. The suggested references are closely related to our work and will be included in the revised version. It is easy and preferred to extract bond disconnection labels from atom mapping. At the same time, our method also works for datasets without atom mapping. Strictly speaking, it is not fully template-free because of the semi-template. However, the semi-template is not a must option for EGAT, and its performance degrades only slightly when removing the semi-template indicator (L143-146).

[1] Shi, Chence, et al. "A Graph to Graphs Framework for Retrosynthesis Prediction." ICML, 2020.

[2] Pensak, David A., et al. "LHASA—Logic and Heuristics Applied to Synthetic Analysis." ACS, 1977. 1-32.


[Meta-Review · NeurIPS 2020]

A paper dealing with an important, topical application. Novel approach and strong empirical results. Reviewers were happy with the author response. The final version should clarify some technicall details and refer more broadly to the recent relevant literature.